# Service Trust and Customer Loyalty in China's Hotel Services: The Causal Role of Commitment

**Jialei Xu**

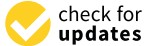



Graduate School of Business Administration, Dankook University, Yongin 16890, Korea; xujia_2862@dankook.ac.kr

**Abstract:** The hotel industry experienced substantial economic losses during the COVID-19 pandemic, and it is only recently that there has been a return to normal. In order to recover those losses, hotel managers are taking competitive actions focused on customer trust as a factor of utmost importance to guarantee customer commitment so that uncertainty and risk can be reduced. In particular, it is necessary to confirm the importance of relationship quality management for target customers in order to secure long-term profits for upscale hotels. Therefore, the purpose of this study is to confirm the antecedents of service trust, the effect of service trust on loyalty through commitment factors, and the moderating effect of length of relationship. For this purpose, a total of 303 questionnaires were collected from Chinese consumers who have experienced upscale hotel service, and empirical analysis was conducted on the data using structural equation model analysis. The results of this study confirmed the importance of trust in front-line employees to build service trust, the different influences of factors of commitment and loyalty, and the moderating effect of length of relationship. Furthermore, efficient customer relationship management can be achieved by classifying the customer's commitment type.

**Keywords:** perceived service quality; service satisfaction; employee trust; service trust; commitment; loyalty

## 1. Introduction

The worldwide COVID-19 pandemic has lasted several years, but through the spread of vaccines and continuous quarantine, China has recently returned to daily life. Both government and private companies are making great efforts to improve the economy, which has been restrained for a while, and maintain qualitative economic activity. In particular, in the case of a service industry that is based on face-to-face contact with customers, such as hotel services, enormous economic losses have been experienced; thus, efforts are being made in various marketing activities to increase the trust and loyalty of target customers in the service. A recent study also stated that in the hotel service industry, which has been badly affected by the spread of COVID-19, marketing managers need to conduct competitive and actionable activities, in particular, marketing strategies, to increase customer trust and loyalty [1]. In previous studies, even if customers are satisfied with the services provided, the hotel service industry cannot guarantee that the customers will revisit a hotel because of the varied and unique services being provided in the industry, intensifying competition among companies [2]. It can be confirmed that compared to building customer loyalty based merely on customer satisfaction, it is more desirable to increase the reliability of the service provided to customers so that customers can commit to the service and increase their loyalty. Therefore, it is important for hotel operators and marketers to manage relationship quality in order to increase the level of trust and commitment of profitable target customers in the service. Trust is the most important factor for organizational sustainability because it is essential for overcoming uncertainty and alleviating risks and anxiety when in crisis; it is also essential in normal

times when service is disrupted [3]. In addition, trust plays an important mediating role in maintaining long-term relationships between customers and service organizations [4–7]. As such, the perceived risk can be very high because customers make decisions about purchasing or using the service before experiencing the service, so service trust is essential in the service industry. Trust is a fundamental factor that is linked to past experiences and is used to forecast future behavior by reducing uncertainty and risk; thereby, it reduces transaction costs and increases efficiency [5]. It can be affirmed that trust in service is of paramount importance in order to guarantee the long-term loyalty of the hospitality company's customers. In research on the antecedents of service trust, perceived service quality [8,9] and overall satisfaction [8,10] are generally regarded as factors affecting the formation of trust in service organizations, and the interaction between customers and employees is a fundamental feature of service because it has unique characteristics that are inseparable from production and consumption [11]. Front-line employee interactions are important to customers' perceived service trust [6,7], perceived service quality, service satisfaction, employee trust, and so on, the various antecedent factors that influence service trust. Additionally, the importance of commitment, along with service trust, has been explained in terms of hotel service; an effective means of guaranteeing the profits of a service company is to maintain a relationship with its customers; hence, customer commitment is an important factor influencing long-term relationship formation [12–14]. Positive beliefs in one another will lead to a commitment to mutual relationships [15,16]. As described above, it will be important to increase customer loyalty, such as their intentions to repurchase and give word-of-mouth recommendations, by increasing the level of service commitment rather than directly affecting customer loyalty in the service industry. In a previous study regarding commitment, it was found that there are two dimensions of commitment: cognitive and affective [17]. Hotel service users evaluate the service provided by the hotel and the service of front-line employees who directly interact with customers on-site; hence, it is necessary to check which service types affect customer service trust and whether service trust affects loyalty through a commitment process. In addition, relationship quality, such as trust and commitment that strengthens the relationship between two parties, can vary depending on the length of relationship, which is a very important factor in the relationship-building process [5,18]. Therefore, it is necessary to confirm the moderating role of length of relationship between service trust and commitment factors; however, no research has been done in this area.

Accordingly, this study aims to identify the antecedents influencing service trust and the relationship between service trust, cognitive commitment, affective commitment, and customer loyalty. Second, the purpose of this study is to examine the moderating role of relationship length on the relationship between service trust, cognitive commitment, and affective commitment.

## 2. Literature Review and Hypothesis

### 2.1. Perceived Service Quality

Perceived service quality refers to the degree to which the service provided matches customer expectations and refers to the degree of disconfirmation between subjective expectations and actual perceptions [19]. In previous studies, service quality measurement is taken from the perspective of consumers; the SERVQUAL model, which includes the five dimensions of tangibility, reliability, responsiveness, assurance, and empathy, is the most widely used model. In the case of hotel services, tangibility refers to what is visible, such as room size, building appearance, and cleanliness; reliability means providing the service requested by the customer and the dependability of the service delivery process; responsiveness refers to the state in which hotel employees are always ready to help customers; assurance refers to creating an atmosphere in which the hotel operates according to certain principles and trusts the behavior of its employees; empathy refers to the degree to which customers can empathize with the services provided by the hotel and its employees. These can be explained as important service quality dimensions for service evaluation, as service

is evaluated through the characteristic of services perceived to form customer attitudes [20]. Previous research has found that perceived service quality plays a very important role in establishing firm performance and long-term profits, so service is considered a factor that can differentiate and maintain a competitive advantage in the market [21]. In addition, when customer service is continuously improved and a high level of service quality is achieved, customers do not show purchase resistance even at premium prices, and the hotel's market share increases in the long run [22]. As described above, customers' perception of service quality is an important concept that can maintain and increase the relationship between customers and service companies; it is necessary to increase the level of service quality recognized by customers in order to maintain profitable existing customers and secure new customers.

### 2.2. Service Satisfaction

The concept of satisfaction was first proposed by Oliver, who initially defined it based on the expectation disconfirmation paradigm, a cognitive evaluation that consumers use by comparing their expectations before using a product or service with the results after the consumption experience [23]. However, in subsequent studies, it has been said that satisfaction consists of emotional elements as well as cognitive elements and is formed through a complex psychological process [24]. Other studies have also suggested that satisfaction has two aspects: transaction-specific and cumulative satisfaction [25–27]. It is a process of comparing expectations for a specific product or service based on the expectation disconfirmation paradigm; the paradigm explains that if the result is greater than the expectation, a positive discrepancy and satisfaction occurs, and, on the contrary, if the result is lower than the expectation, negative discrepancy and dissatisfaction are caused [24,26]. On the other hand, with regard to service situation satisfaction, Hansemark and Albinsson [28] described an expectation disconfirmation paradigm, stating that satisfaction with a contracted service is an emotional response to the difference between what the customer expects and the satisfied needs, whereas cumulative satisfaction is the overall evaluation of a service that occurs over time [25]. In a follow-up study, it was shown that the two types of satisfaction have complementary characteristics, where transaction-specific satisfaction tracks customer responses to changes in service performance while cumulative satisfaction focuses more on understanding the customer–firm relationship over time [25]. However, cumulative satisfaction is used more than transaction-specific satisfaction in predicting customer behavior [29]. In a hotel service study, customers preferred to measure overall satisfaction when making decisions based on their overall experience with the service [26]. Therefore, in this study, service satisfaction is defined as the overall evaluation of the customer's service purchase or experience, which also reflects the fulfillment of the customer's needs.

### 2.3. Customer Trust

In previous studies, it has been found that consumers understand the other party through their belief in the reliability and integrity of the supplier; thus, trust is regarded as a marketing element that plays a key role in creating and maintaining long-term relationships with customers [30]. In particular, trust plays a role in reducing customer anxiety and can reduce pre-purchase steps such as information retrieval in the purchasing decision-making process [4]. Additionally, building trust by maintaining long-term relationships with customers is the most basic premise for service companies to secure stable profits. In particular, with regard to trust in hotel services, there are two factors that customers consider to be as important as the capabilities of the hotel: credibility trust, such as infrastructure, employee behavior, expected information before check-in, and services to be provided, and benevolence trust, where the customer's well-being and benefit should be kept in mind [31,32]. In another study, trust is divided into two types: trust in front-line employees and trust in corporate management policies [33]. In addition, although trust in the service organization is important because of the inseparability of production and consumption in

high-involvement services, the interaction between customer and employee is the basis of service, explaining the importance of trust when it comes to interactions with front-line employees [11]. Service employee trust can be evaluated through convenience, trustworthiness, honesty, and continuous relationship intention [34]. In particular, in the field of upscale hotel service, employee trust is necessary to reduce uncertainty [35]. Therefore, in order to provide high customization in upscale hotel service management, it is necessary to maintain trust relationships in two aspects: front-line employees and service organizations in contact with customers.

### 2.4. Customer Commitment

Commitment refers to the maximum effort to maintain a relationship with customers [12]. Although the conceptual definition of commitment is defined in various fields such as psychology, organizational behavior, and marketing, it is a common view that it includes the psychological state and motivational behavior [36]. The essence of commitment between customers and the organization implies two aspects: the feeling of identifying with the consumer and when individuals maintain a relationship with the organization because of their needs, obligations, or desires [37]. In terms of hotel service, customer commitment is a very important factor because long-term relationships with highly profitable customers directly affect competitiveness and long-term profitability [10,38]. In prior research, commitment has been classified into three components: affective commitment refers to a state of attachment that prefers an exchange partner or has a sense of identity; normative commitment is the effort to maintain a relationship with the exchange partner because of a feeling of obligation; calculative commitment can be seen as the need to maintain a relationship due to the lack of alternatives or the expected transition cost [16,36,39]. It can be confirmed that these dimensions of commitment are ultimately classified into emotional, rational, and moral dimensions [40]. However, it was argued that only two dimensions of commitment are necessary in the field of marketing: rational bond (cognitive aspect) and affective bond (affective aspect) [17].

### 2.5. Customer Loyalty

Loyalty is an outcome variable of relationship marketing and has an important role in the company's sustainable profitability based on its relationship with customers. In previous studies, loyalty is defined as the continued use of or commitment to a preferred service in the future; it has four stages: cognitive, affective, conative, and action [24], which, in later studies, is mostly presented in two dimensions, such as loyalty in the attitude dimension and loyalty in the behavioral dimension. Attitudinal loyalty includes the consumer's affective viewpoint and positive word-of-mouth about a specific product or service rather than a competitor's, whereas behavioral loyalty includes the repurchase of the same product or service or the probability of purchase [41]. In the hotel service field, there is a need to comprehensively classify customer loyalty into attitude loyalty and behavioral loyalty [14]. Therefore, loyalty is expressed as attitudinal loyalty and behavioral loyalty in this study to evaluate the favorability of hotel services and the willingness to use upscale hotel services over a long period of time.

### 2.6. Length of Relationship

In marketing research, the concept of length of relationship refers to the length of time a customer has been with a service provider. Previous studies suggest that there is a difference between younger customers and older customers in the relationship formation process [5,18], suggesting a moderating effect according to the relationship group; however, the difference according to the relationship period has not been explained yet. In previous studies on length of relationship, past-oriented people often repeat past behaviors because they prefer to maintain favorable experiences and want to maintain established relationships, suggesting that there is a positive relationship between corporate reputation and loyalty [42]. However, long-term buyer–seller relationships require more frequent

interactions than long-term bilateral relationships, and the frequent interactions can help buyers obtain important information that may weaken the positive effects and make it easy to be negatively influenced, explaining the effect of different relationship periods [43]. In other studies, length of relationship had little effect on perceived value, customer loyalty, trust, and purchase intention [44,45]. However, studies on the difference between trust and commitment factors according to the relationship period in upscale hotel services have not been proposed yet.

### 2.7. Hypothesis and Research Model

From the perspective of the company, customer trust emphasizes the importance and meaning of trust to promote profitable customer relationships, helping the company differentiate itself from its competitors in terms of service. The way to build stable and trustworthy relationships with customers is to provide quality service that sets the company apart from its competitors in a mature market [9]. Trust is an enduring belief that in the relationship between the customer and the service provider, customers are convinced that the service provider can satisfy their needs [8]. Perceived service quality has a positive effect on customer trust in various service industries, not limited to the hotel industry [7–9]. When there is perceived uncertainty in service provider–customer relationships, trust can be increased by satisfying customer perceptions by providing better service quality. Therefore, the following research hypothesis is proposed.

**Hypothesis (H1):** *Perceived service quality has a positive (+) effect on service trust.*

In a previous study, Cronin [6] and Yim et al. [7] stated that perceived service quality and customer satisfaction are antecedent variables of consumers' attitudes toward service providers, which, in turn, affect purchase intentions; thus, customer trust in service providers explains the mediating role. In order for customers who are overall satisfied with the service organization to be loyal, it can be confirmed that the trust of the customers in the interaction relationship with the service provider is important. Furthermore, satisfaction is generally a source of trust-building, and it is explained that satisfaction has a positive relationship with trust by providing a service to consumers who have already had a satisfying experience [7]. In a study of 4- and 5-star hotels, it was confirmed that the essential antecedent factor of trust is satisfaction, and, through this, it was confirmed that the trust of consumers who are overall satisfied with the hotel service can increase [8,10]. Therefore, the following research hypothesis is proposed.

**Hypothesis (H2):** *Service satisfaction has a positive (+) effect on service trust.*

Building trust between service companies and customers is the basic goal of service companies, and for this, building trust between customers and front-line employees must precede [46]. In particular, in high-contact services such as tourism services with service employees in the service delivery process, the interaction in the delivery process affects the level of credence, so it is necessary to increase the level of interaction evaluated by consumers [10]. In the case of hotel service, the role of employee trust is important to improve relationship quality [34,47]. In addition, in a study on the relationship between customer–employee trust and customer–corporate trust, it was confirmed that employee trust had a significant effect on corporate trust as an antecedent factor [7]. Therefore, in order to build trust in the hotel industry, it will be necessary to form trusted relationships with employees who come into contact with customers, above all else. Therefore, the following research hypothesis is proposed.

**Hypothesis (H3):** *Employee trust has a positive (+) effect on service trust.*

Trust-commitment theory is generally seen to have an important mediating role between long-term institutions. To date, many marketing studies have described the relationship between trust and commitment [15,16,30,48]. Looking at the relationship between trust and commitment in the service environment, trust creates a sense of connectedness

and empathy in the psychological process between service providers and service buyers; it strives to form and maintain relationships for mutual benefit [16]. Hennig-Thurau and Klee [17] stated that trust can promote commitment by two methods: efficiency and social needs. An increase in relationship efficiency accompanies an increase in customer profits, which, in turn, promotes the customer's cognitive commitment to the relationship. For example, if you have relationship efficiency with the employees through predictability and immediate responsiveness (among the characteristics of trust), you will receive excellent services by further customizing service contacts. Previous studies have also shown that in the case of low trust, consumers tend to supervise the employee's actions and will eventually calculate whether continuing or terminating a relationship is a cost or benefit [15]. Therefore, if customer trust is high, it is difficult to deviate from the relationship with the service provider, and, rationally, customers will strive to maintain the long-term relationship because it is possible to recognize mutually beneficial facts. Therefore, the following research hypothesis is proposed.

**Hypothesis (H4):** *Service trust has a positive (+) effect on cognitive commitment.*

Previous studies have explained that trust deals with the core social needs of customers, and their satisfaction leads to affective commitment in the relationship [17,48]. In addition, service employees can identify customer needs to provide satisfying services and form intimate social bonds [35]. When a service company satisfies the social needs of customers, it forms an emotional state of preference or identity and, thus, forms a relationship. Moreover, customers are more aware of the connectedness and identification of trusted counterparts; the more they trust their suppliers, the more likely they are to feel the urge to continue with the relationship [48]. When trust increases, customers tend to prefer the same service providers and enjoy their services, which can increase the likelihood that the continuity of the relationship will be maintained. In addition, trust, as an important parameter of affective commitment in long-term relationships, has a positive effect on affective commitment [16]. Therefore, the following research hypothesis is proposed.

**Hypothesis (H5):** *Service trust has a positive (+) effect on affective commitment.*

In previous studies on the relationship between commitment and loyalty in the hotel service industry, it has been explained that commitment directly affects loyalty in the relationship between service providers and consumers; the desire to continue the relationship and the will to maintain the relationship increase customer loyalty [10,14]. In addition, another study explained the effect on customer loyalty, saying that consumers have a tendency to keep suppliers without changing them in order to avoid switching costs or other costs that are due to economics and scarcity [16]. However, the higher the level of commitment, the more positive the attitude toward the favorable group because it meets the needs of target customers by providing high-level services with a high level of individual customization and predictability of behavior in the case of upscale service. In other studies, it was shown that if consumers wish to leave the service provider they use and move on to another service provider, a high level of switching cost will occur; hence, they tend to continue using their current provider. In this case, it is said that they become cognitively immersed in the service and will give positive evaluations to those around them [36]. Based on the preceding studies, the following research hypothesis is proposed.

**Hypothesis (H6):** *Cognitive commitment has a positive (+) effect on attitudinal loyalty.*

In previous studies, it was reported that consumers become deeply committed in the future to rebuy or repatronize a preferred product or service [49]. It can be confirmed that the behavioral level of loyalty, such as the repurchase intention, is affected by the customer's commitment. In other studies, it is also explained that commitment, which is the basis of consumer behavioral loyalty, is a process in which a psychological bond is formed through internalized and justified switching costs, and the commitment formed by service evaluation has a significant effect on the behavioral loyalty of consumers [36].

In addition, if customers try to switch the hotel service they have used, special services, accumulated points, upgrades, and other benefits are lost, so a high level of switching cost occurs when switching services [50]. Conversely, by being given various benefits through the formation of a long-term favorable relationship, the motivation to stay in the relationship is increased, and behavioral loyalty is also induced [51]. Therefore, consumers will perceive high switching costs when receiving special treatment and unique benefits through a high level of service quality in their preferred hotel; hence, they can continue to use the same hotel. A relatively recent study also explained that cognitive commitment has a positive effect on behavioral loyalty because commitment based on economic and rational values in hotel services is positively related to behavioral loyalty [50,52]. Therefore, the following research hypothesis is proposed.

**Hypothesis (H7):** *Cognitive commitment has a positive (+) effect on behavioral loyalty.*

Affective commitment is viewed as an emotional connection that maintains a positive relationship between the customer and the firm [30]. In previous studies, factors included in affective commitment, such as happiness, belonging, recognition, and perception of attachment, were closely related to loyalty [53], and affective commitment was identified as a key concept underlying loyalty. Affective commitment is the main concept underlying loyalty, and it is achieved through identification and attachment [54]; hence, the effect of affective commitment on loyalty is explained. In addition, the identification that customers feel toward the service firm leads to the transmission of positive emotions to others [55]. In particular, in hotel service research, when consumers focus on various benefits rather than excellent service quality, this does not necessarily increase customer loyalty; on the other hand, customers with a high affective commitment will not only consider the brand as their first choice but will also promote the brand to their friends and colleagues. It was explained that the higher the affective commitment, the greater the likelihood of promoting the brand and influencing loyalty [56]. Additionally, in other studies, high affective commitment had a strong effect on consumers' attitudinal loyalty [57]. Therefore, the following research hypothesis is proposed

**Hypothesis (H8):** *Affective commitment has a positive (+) effect on attitudinal loyalty.*

Affective commitment increases loyalty through the emotional bonds that develop through the personal involvement or reciprocity that consumers have with the company [16]. Evanschitzky et al. [57] demonstrated the superiority of affective commitment in predicting behavioral loyalty in the marketing literature, and research on service customer commitment and response by Jones et al. [36] confirmed that affective commitment is a major dimension of customer behavioral response. In addition, it was confirmed that there is a positive relationship between customer commitment and a sense of belonging and repeat purchase behavior [13]; in other studies, affective commitment was a strong predictor of purchase intention and an important factor in strengthening the relationship [58]. In the study of luxury hotel services, a low attachment to hotels increases the risk of leaving, while customers with a high attachment have a continuous intention to visit their preferred hotels [14,35]. Therefore, the following research hypothesis is proposed.

**Hypothesis (H9):** *Affective commitment has a positive (+) effect on behavioral loyalty.*

When evaluating the relationship between attitude and behavior, behavioral intention can be an antecedent factor of actual behavior as the basis for the theory of reasoned action [59]. This theory explains that people act on what they intend to do and do not act on what they do not intend to do. Attitudinal loyalty emphasizes the customer's strong positive attitude toward the seller or brand, and the strength of this attitude motivates the customer to defend the brand with a strong opinion in front of others [60]. The influence of customer recommendation, advocacy, positive purchase intention, and attitude on behavior was verified [10]. Additionally, attitudinal loyalty has a positive effect on behavioral loyalty in luxury hotel services [35]. Therefore, the following research hypothesis is proposed.

**Hypothesis (H10):** *Attitudinal loyalty has a positive (+) effect on behavioral loyalty.*

The level of customers' interaction with service providers will be different according to their time of service use as well as the quality and quantity of information they have collected, indicating that there is a difference in the process of consumers' attitude formation [5]. A study by Wang and Wu [61] explains the need for buyers to maintain relationships with suppliers in order to achieve their desired goals. For example, services provided by service providers in a stable and reliable manner may strengthen the older customer–provider relationship and result in higher conversion costs, but in the younger exchange relationships, customers may not have sufficient knowledge of the company, so they collect information through advertisements, word-of-mouth, and other service contacts. Additionally, corporations identify customer needs and opportunities to build strong relationships, suggesting that an increase in the number of satisfactory business–customer interactions leads to greater trust and greater commitment to the relationship [30]. Specifically, as the service length of the relationship develops, trust characteristics such as the customization and behavioral predictability of services have an effect on commitment [5]. Additionally, Bove and Johnson [51] argued that the longer the relationship between the customer and the service provider, the stronger the role it plays because it is self-centered and justifies the positive efforts of the corporation and the choice of the consumer. Therefore, based on the above research, the following research hypothesis is proposed.

**Hypothesis (H11):** *The effect of trust on cognitive commitment differs according to the length of the relationship.*

In terms of social exchange characteristics, the frequency and intensity of contact allow customers to form an impression of the firm's relationship efforts and benefits [62]. Additionally, with longer relationships, customers can have a broader knowledge structure that can more accurately evaluate a firm's relationship efforts [63]. Morgan and Hunt [30] argued that a satisfactory interaction of relational bonds results in a longer relationship duration as the number of interactions increases. When a satisfactory relationship with a service company is formed, the bond is strengthened, and the longer the relationship of trust in the service provided, the higher the level of affective commitment to the service. Therefore, based on the above research, the following research hypothesis is proposed.

**Hypothesis (H12):** *The effect of trust on emotional commitment differs according to the length of the relationship.*

As with the research hypothesis presented above, this study identifies perceived service quality, service satisfaction, and employee trust in the upscale hotel service context as the antecedent factors of service trust. In addition, in the process of customer-firm long-term relationship formation, this study confirms whether service trust builds customer loyalty through two dimensions of commitment, cognitive and affective dimension. This study also verifies the moderating effect of length of relationship in the process of service trust affecting cognitive and affective commitment. The research model is shown in Figure 1.

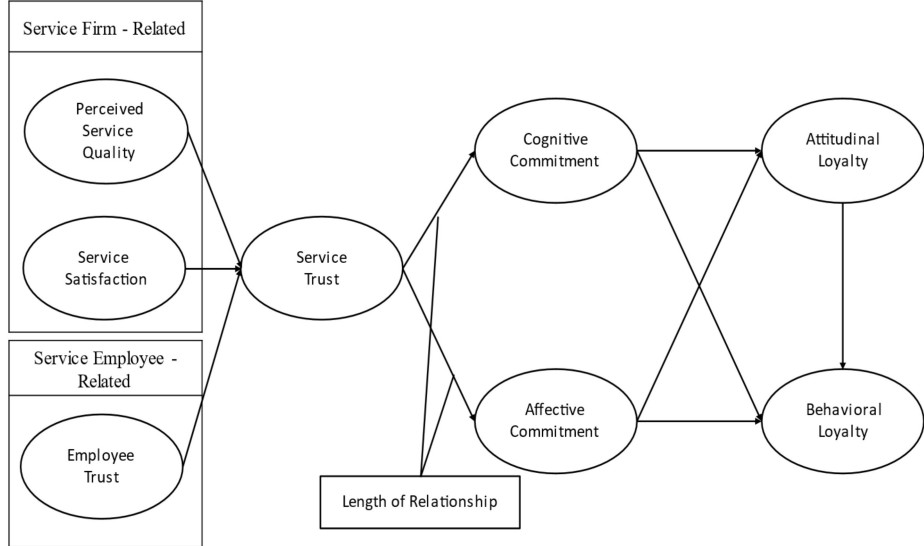

**Figure 1.** Research model.

## 3. Methodology

### 3.1. Measurement and Operational Definition of Variables

The contents of the constructs presented in the research model of this study were modified to fit the contents of this study based on the related previous studies. A multi-item scale was developed to identify the domain of the construct and to help the comprehensive understanding of it. The measurement used in this study is a 7-point Likert scale consisting of '1. Strongly disagree' to '7. Strongly agree'. Table 1 shows the measurement items and operational definitions of the constructs.

**Table 1.** Items and operational definition of constructs.

| Constructs | | Items | Operation Definition | Research |
|---|---|---|---|---|
| Perceived Service Quality | PQ1 | The hotel was visually appealing. | The degree of discrepancy between the subjective expectations of customers using hotel services and their actual perceptions. | [19,20] |
| | PQ2 | My reservation was handled efficiently. | | |
| | PQ3 | Employees responded promptly to my requests. | | |
| | PQ4 | The hotel provided a safe environment | | |
| | PQ5 | Charges on my account were clearly explained. | | |
| Service Satisfaction | SS1 | Overall satisfied with the content and results of the hotel service. | The degree to which the service generally satisfied the customer's needs after the customer using the hotel service experiences the service. | [26,64] |
| | SS2 | Overall satisfied with hotel service execution process. | | |
| | SS3 | Overall satisfied with the hotel service use environment. | | |
| | SS4 | Overall satisfied with the hotel service. | | |
| | SS5 | Overall satisfied with the hotel service against expectations after use. | | |
| Employee Trust | ET1 | The hotel employee kept promises. | The degree of customer belief that hotel service personnel will perform positive service behaviors. | [34,47] |
| | ET2 | The hotel employee was sincere. | | |
| | ET3 | The hotel employee was reliable. | | |
| | ET4 | The hotel employee was honest. | | |
| | ET5 | The hotel employee put customers' interests first. | | |
| Service Trust | ST1 | The hotel service is reliable. | The extent to which hotel services are believed to provide reliable service. | [14,31] |
| | ST2 | The hotel service protects customers' personal information and maintain transactions. | | |
| | ST3 | The hotel service the service is always honest. | | |
| | ST4 | The hotel service keeping promises with customers. | | |
| | ST5 | Other customers will also trust the hotel service. | | |

**Table 1.** *Cont.*

| Constructs | | Items | Operation Definition | Research |
|---|---|---|---|---|
| Cognitive Commitment | CC1 | I don't think the experience here can be found anywhere else. | The degree to which one perceives and strives for the need to maintain a relationship with the hotel used because there is no alternative or the conversion cost is expected to occur. | [17,36] |
| | CC2 | I believe that this hotel provides a good service that cannot be found anywhere else. | | |
| | CC3 | I like this hotel and will continue to use it. | | |
| | CC4 | I would like to continue to maintain a relationship with this hotel in the future. | | |
| | CC5 | Relations with this hotel will be well maintained in the future. | | |
| Affective Commitment | AC1 | I feel "emotionally connected" to this hotel. | Degree of attachment to hotels that prefer hotel services and feel a sense of identity. | [35,53] |
| | AC2 | I have a sense of attachment to this hotel. | | |
| | AC3 | The warmth of the staff in this hotel makes me pleased. | | |
| | AC4 | I enjoy visiting this hotel. | | |
| | AC5 | Although there are other hotel options, I still like going to this hotel. | | |
| Attitudinal Loyalty | AL1 | I have pleasant feelings in this hotel. | The degree to which customers have an affective view and favorable attitude toward hotel services. | [14,65] |
| | AL2 | I like this hotel the most. | | |
| | AL3 | I have a favorable attitude towards this hotel. | | |
| | AL4 | I would like to tell other good thing about this hotel. | | |
| | AL5 | I have a more favorable attitude to this hotel than anywhere else. | | |
| Behavioral Loyalty | BL1 | I will recommend this hotel to friends and relatives. | Intentions and behaviors of customers repeatedly using and recommending hotel services. | [14,66] |
| | BL2 | I will remain using this hotel. | | |
| | BL3 | If I have a chance, I want to visit this hotel again. | | |
| | BL4 | If I had knowledge of this hotel before, I would have used it. | | |

*3.2. Research Design*

To verify the research hypothesis presented in this study, data were collected using a questionnaire. The questionnaire consisted of perceived service quality, service satisfaction, employee trust, service trust, cognitive commitment, affective commitment, attitudinal loyalty, and behavioral loyalty as major variables. As questions to categorize the data, the type of hotel, purchased service, and purchase period were included. To confirm that the sample of this study describes the population characteristics of China's upscale hotel customers, questions on demographic characteristics such as gender, age, education level, income level, and occupation were incorporated. In general, the selection of survey subjects should be based on four criteria, such as the research subject, scope, time, and sample unit, so the sample was selected based on these criteria for consumers who have experience in using upscale hotel services in China. A convenience sampling method was used to target consumers who have experienced upscale hotel services in China, and 303 pieces of data were collected. Based on a questionnaire from previous studies, the contents of the measurement items were modified to fit the purpose and context of this study. The survey period was from 15 January to 15 March 2022.

*3.3. Research Method*

The data collected for this study were analyzed using SPSS 21.0 and AMOS 20.0 statistical software. To describe the population characteristics of the sample, frequency analysis was performed. KMO and Bartlett's test were used to confirm the validity and reliability of the data to be used for factor analysis. To verify the convergent validity of the construct, confirmatory factor analysis (CFA) was performed, and discriminant validity was confirmed by comparing the value derived from the correlation analysis with the AVE value. Reliability was confirmed with Cronbach's $\alpha$ value by performing internal consistency analysis. Basic statistics such as mean and standard deviation (S.D.) were

analyzed to verify the collected data. Lastly, to verify the model fit and research hypotheses of the study, structural equation modeling (SEM) was performed.

## 4. Empirical Analysis

### 4.1. Analysis of Respondent's Demographics and Data Characteristics

In this study, 320 questionnaires were collected from customers who had used luxury hotel services in China. A total of 303 copies (94.7%) was used for empirical analysis, excluding 17 questionnaires that did not respond faithfully or responded inappropriately. As a result of analyzing the characteristics of the data, the hotels were Hilton (77, 25.4%), Sheraton (73, 24.1%), Shangri-La (34, 11.2%), Marriott (37, 12.2%), Four Seasons (50, 16.5%), and others (6, 2%). The services used were accommodation (132, 43.6%), restaurant (72, 23.8%), meeting room service (25, 8.3%), lounge service (32, 10.6%), health service (33, 10.9%), and other services (9, 7.3%). Table 2 shows the demographic characteristics analysis results of the respondents.

**Table 2.** Results of respondents' demographic characteristics analysis.

| Variables | Attributes | No. (%) |
|---|---|---|
| Gender | Male | 143 (47.2) |
| | Female | 160 (52.8) |
| Age | Below 20 years old | 8 (2.8) |
| | 20–29 years old | 54 (17.8) |
| | 30–39 years old | 105 (34.7) |
| | 40–49 years old | 72 (23.8) |
| | 50–59 years old | 50 (16.5) |
| | Over 60 years old | 14 (4.6) |
| Education | High School or Below | 60 (19.8) |
| | Undergraduate | 188 (62.0) |
| | Graduate or higher | 53 (18.2) |
| Incomes | Below RMB 8000 | 91 (30.7) |
| | RMB 8000~16,000 | 121 (39.9) |
| | Over RMB 16,000 | 89 (29.4) |
| Occupation | Student | 18 (5.9) |
| | Private company's employee | 130 (42.9) |
| | Government official | 50 (16.5) |
| | Full-time housewife | 21 (6.9) |
| | Business owner | 63 (20.8) |
| | Other | 21 (6.9) |

Note: The questions only had one possible answer.

### 4.2. Validity and Reliability Analysis

To check the correlation between variables, KMO and Bartlett's test were performed. The KMO measurement value was 0.927; the Bartlett test analysis, showing the significance of all correlation values, was found to be significant ($p < 0.000$), so it was finally confirmed to be suitable for factor analysis. Confirmatory factor analysis (CFA) was evaluated with GFI, NFI, CFI, RMR, and RMSEA fit indices. The analysis results were $x^2$ (294.974), $df$ (224), $p$ (.000), $x^2/df$ (1.317), GFI (0.926), NFI (0.939), CFI (0.984), RMR (0.046), and RMSEA (0.032), and the fit was confirmed. In addition, all construct reliability (C.R.) was found to be higher than the standard value of 0.7, and all mean variance extraction values (AVEs) were shown to be higher than the standard value of 0.5, confirming the convergent validity of the data [67]. Additionally, the result of reliability analysis by internal consistency analysis was higher than 0.7 in all variables, confirming the reliability of the data. Table 3 shows the analysis results.

Discriminant validity was analyzed by converting the $r$ value derived through correlation analysis into $r^2$ and comparing it with the AVE value. As a result of the analysis, the

significance level ($p$ <0.05) showed a significance probability (0.000) that was significant in all variables [67]. Additionally, the AVE value was higher, confirming the discriminant validity between variables. Table 4 shows the analysis results.

**Table 3.** Results of confirmatory factor analysis, reliability analysis, and KMO and Bartlett's test.

| Construct | CFA Before | CFA After | Items | Estimate | Standardized Regression Weights | C.R | p | AVE | C.R. | α |
|---|---|---|---|---|---|---|---|---|---|---|
| Perceived Service Quality | 5 | 3 | SQ1<br>SQ2<br>SQ4 | 1.000<br>0.887<br>0.892 | 0.817<br>0.840<br>0.895 | <br>18.340<br>17.601 | <br>0.000 ***<br>0.000 *** | 0.725 | 0.883 | 0.886 |
| Service Satisfaction | 5 | 3 | SS3<br>SS4<br>SS5 | 1.000<br>1.019<br>0.984 | 0.827<br>0.815<br>0.809 | <br>15.190<br>15.076 | <br>0.000 ***<br>0.000 *** | 0.668 | 0.799 | 0.858 |
| Employee Trust | 5 | 3 | ST1<br>ST3<br>ST5 | 1.000<br>1.020<br>0.984 | 0.808<br>0.819<br>0.744 | <br>13.255<br>13.407 | <br>0.000 ***<br>0.000 *** | 0.626 | 0.824 | 0.827 |
| Service Trust | 5 | 3 | SA2<br>SA3<br>SA4 | 1.000<br>0.984<br>1.048 | 0.833<br>0.815<br>0.809 | <br>14.878<br>15.181 | <br>0.000 ***<br>0.000 *** | 0.671 | 0.857 | 0.859 |
| Cognitive Commitment | 5 | 3 | CC1<br>CC2<br>CC3 | 1.000<br>1.018<br>1.076 | 0.839<br>0.878<br>0.874 | <br>18.515<br>18.414 | <br>0.000 ***<br>0.000 *** | 0.746 | 0.872 | 0.898 |
| Affective Commitment | 5 | 3 | AC1<br>AC3<br>AC5 | 1.000<br>0.860<br>1.064 | 0.881<br>0.797<br>0.865 | <br>16.255<br>18.141 | <br>0.000 ***<br>0.000 *** | 0.720 | 0.878 | 0.884 |
| Attitudinal Loyalty | 5 | 3 | AL2<br>AL3<br>AL4 | 1.000<br>1.027<br>0.970 | 0.849<br>0.827<br>0.860 | <br>16.679<br>17.159 | <br>0.000 ***<br>0.000 *** | 0.715 | 0.852 | 0.884 |
| Behavioral Loyalty | 4 | 3 | BL2<br>BL3<br>BL4 | 1.000<br>1.027<br>0.970 | 0.797<br>0.915<br>0.828 | <br>17.266<br>15.816 | <br>0.000 ***<br>0.000 *** | 0.719 | 0.851 | 0.881 |

$x^2/df = 294.974/224 = 1.317$, $p = 0.000$, RMR = 0.046, GFI = 0.926, NFI = 0.939, CFI = 0.984, RMSEA = 0.032;

KMO = 0.927, $p = 0.000$, approx. chi-square = 9247.995, $df = 820$

Note: *** $p < 0.001$.

**Table 4.** Result of discriminant validity analysis.

| AVE | 1 | 2 | 3 | 4 | 5 | 6 | 7 | 8 |
|---|---|---|---|---|---|---|---|---|
| 1 | 0.725 | | | | | | | |
| 2 | 0.473 ***<br>(0.224) | 0.668 | | | | | | |
| 3 | 0.531 ***<br>(0.282) | 0.601 ***<br>(0.361) | 0.626 | | | | | |
| 4 | 0.452 ***<br>(0.204) | 0.367 ***<br>(0.135) | 0.502 ***<br>(0.252) | 0.671 | | | | |
| 5 | 0.346 ***<br>(0.120) | 0.295 ***<br>(0.087) | 0.423 ***<br>(0.179) | 0.526 ***<br>(0.276) | 0.746 | | | |
| 6 | 0.377 ***<br>(0.142) | 0.388 ***<br>(0.151) | 0.405 ***<br>(0.164) | 0.294 ***<br>(.086) | 0.301 ***<br>(0.091) | 0.720 | | |
| 7 | 0.292 ***<br>(0.085) | 0.304***<br>(0.092) | 0.312 ***<br>(0.097) | 0.383 ***<br>(0.147) | 0.585 ***<br>(0.342) | 0.255 ***<br>(0.065) | 0.715 | |
| 8 | 0.420 ***<br>(0.176) | 0.332 ***<br>(0.110) | 0.359 ***<br>(0.129) | 0.318 ***<br>(0.101) | 0.290 ***<br>(0.084) | 0.318 ***<br>(0.101) | 0.347 ***<br>(0.120) | 0.719 |
| Mean | 4.641 | 4.634 | 4.430 | 4.661 | 4.734 | 4.735 | 4.557 | 4.701 |
| S.D. | 0.925 | 1.086 | 0.889 | 0.895 | 1.041 | 0.943 | 1.044 | 1.028 |

Note: *** $p < 0.001$, ( ) = $r^2$. Service quality = 1, service satisfaction = 2, employee trust = 3, service trust = 4, cognitive commitment = 5, affective commitment = 6, attitudinal loyalty = 7, behavioral loyalty = 8.

### 4.3. Hypothesis Test

#### 4.3.1. Analysis of Model Fit and Hypothesis Test

Results of analysis to confirm the fit of the research model presented in this study were $p$ (0.000), $x^2/df$ (389.235/239 = 1.629), GFI (0.905), NFI (0.921), CFI (0.968), and RMSEA (0.046), confirming the fitness of the research model. Accordingly, the research hypothesis proposed in this study was confirmed. As a result of the analysis, perceived service quality (0.269, $p < 0.05$) and employee trust (0.474, $p < 0.05$) had a statistically significant effect on service trust. However, service satisfaction ($-0.013$, $p = 0.855$) was not statistically significant. Service trust had a statistically significant effect on cognitive commitment (0.794, $p < 0.05$) and affective commitment (0.464, $p < 0.05$), and cognitive commitment had a significant effect on both attitudinal loyalty (0.225, $p < 0.05$) and behavioral loyalty (0.305, $p < 0.05$). However, affective commitment had a direct effect on attitudinal loyalty (0.632, $p < 0.05$) but no effect on behavioral loyalty. Therefore, hypotheses H1, H3, H4, H5, H6, H7, H8, and H10 are supported, whereas hypotheses H2 and H9 are rejected. Table 5 and Figure 2 show the analysis results.

**Table 5.** Results of hypothesis testing.

| Hypothesis | Estimate | S.E. | C. R | $p$ | Results |
|---|---|---|---|---|---|
| 1 | 0.269 | 0.073 | 3.702 | *** | Supported |
| 2 | −0.013 | 0.072 | −0.183 | 0.855 | Rejected |
| 3 | 0.474 | 0.101 | 4.707 | *** | Supported |
| 4 | 0.794 | 0.079 | 10.009 | *** | Supported |
| 5 | 0.464 | 0.072 | 6.413 | *** | Supported |
| 6 | 0.225 | 0.059 | 3.848 | *** | Supported |
| 7 | 0.305 | 0.067 | 4.587 | *** | Supported |
| 8 | 0.632 | 0.058 | 10.942 | *** | Supported |
| 9 | −0.029 | 0.058 | −0.503 | 0.615 | Rejected |
| 10 | 0.234 | 0.058 | 4.029 | *** | Supported |

$x^2/df$ (389.235/239) = 1.629, GFI (0.905), NFI (0.921), CFI (0.968), TLI (0.963), RMSEA (0.046)

Note: *** $p < 0.001$.

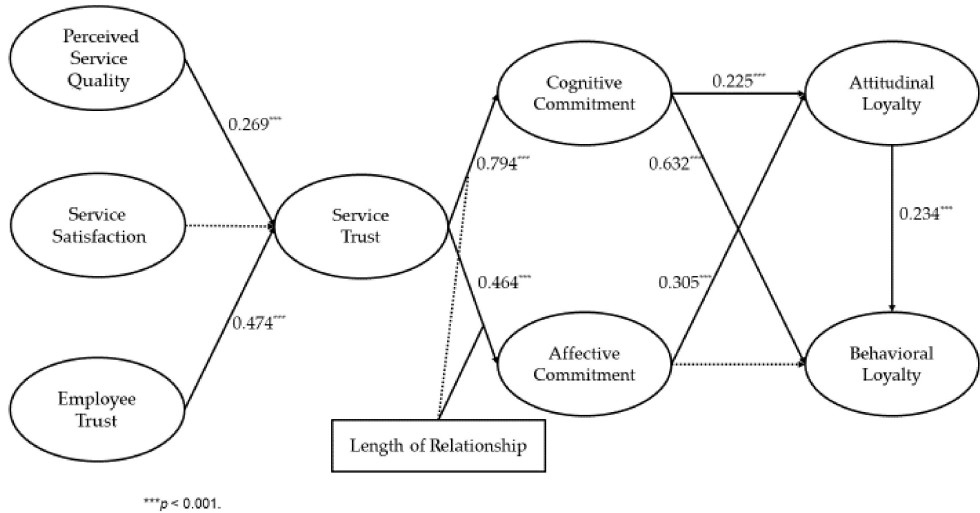

***$p < 0.001$.

**Figure 2.** Final model and path coefficients.

#### 4.3.2. Analysis of the Moderating Effect

In order to confirm the moderating effect of the length of relationship in each relationship between service trust, cognitive commitment, and affective commitment, based on the median (2.00), the group was divided into a low group (n = 140) of 1~2 and a high group (n = 163) of 2~6. The analysis confirms the research hypothesis by checking whether the path coefficients for each group correspond with the research hypothesis and direction and

by verifying the difference between the free model and the constrained model. First, in order to confirm the direction of the moderating effect, the coefficient of the high group should be larger than that of the low group. As a result of the analysis, both hypothesis H11 and hypothesis H12 showed that the high group was larger than the low group, so the direction was consistent. Additionally, the result confirming the difference in chi-square () between the free model and the constrained model, the $\Delta x^2$ of 0.005 in H11, was lower than the standard value of 3.84, which was not statistically significant, whereas the $\Delta x^2$ in H12 was 5.101, which was statistically higher than the standard value of 3.84. Therefore, hypothesis H11 is rejected, while hypothesis H12 is supported. Table 6 shows the analysis results.

**Table 6.** Results of moderating effect analysis.

| Hypotheses | High Group | | | Low Group | | | Free Model | | Constrained Model | | $\Delta x^2$ | Result |
|---|---|---|---|---|---|---|---|---|---|---|---|---|
| | Estimate | t | p | Estimate | t | p | df | $x^2$ | df | $x^2$ | | |
| H11 | 0.840 | 5.953 | *** | 0.828 | 8.661 | *** | 478 | 634.690 | 479 | 634.695 | 0.005 | Rejected |
| H12 | 0.795 | 6.339 | *** | 0.465 | 5.402 | *** | | | | 639.791 | 5.101 | Supported |

Note: *** $p < 0.001$.

## 5. Conclusions

### 5.1. Theoretical Implications

This study has found that perceived service quality and employee trust have a positive effect on service trust, whereas service satisfaction does not. Service trust had a positive effect on cognitive and affective commitment. Cognitive commitment had a positive effect on attitudinal loyalty and behavioral loyalty. Affective commitment had a positive effect on attitudinal loyalty but did not affect behavioral loyalty. Attitudinal loyalty had a positive effect on behavioral loyalty. Results confirming the moderating role of relationship length showed that relationship length had no moderating role in the relationship between service trust and cognitive commitment but confirmed a moderating role in the relationship between service trust and affective commitment.

The theoretical implications of the results of this study are as follows. First, it was confirmed that the antecedent factors affecting service trust were discriminatory at the service company level and the service employee level in hotel services, respectively. In other words, perceived service quality as an antecedent factor related to service companies and employee service as a factor related to service employees respectively affected service trust. However, the employee-related antecedent (employee trust) and the company-related antecedent (perceived service quality) have a different level of influence: employee trust has a greater impact on service trust than perceived service quality. This is because a service company that provides excellent quality can promote stability and reliability by satisfying customer needs, differentiating itself from competitors, and recognizing uniqueness [9]. It is possible to check the effect of perceived service quality on service trust. In addition, the active and positive service behavior of front-line employees when interacting with customers in the service industry reduces the perceived risk to customers and increases the level of trust in the service [7,8]. In this study, it was confirmed that employee trust had a high level of influence on service trust, so the results of all previous studies and research were consistent. However, comparing the degree of influence confirmed that the trust in front-line employees interacting at service points (and not the perceived service quality associated with the service entity) leads directly to overall trust in the service and deduces the interaction behavior of front-line employees at the moments of truth. Second, this study confirmed that when service trust affects affective commitment, it does not directly affect customer behavior (such as purchase behavior) but indirectly affects behavioral loyalty through attitudinal loyalty. Additionally, if customers had low affective commitment, there was a risk of losing customers because competitors could easily imitate the service;

hence, that loyalty can be increased by satisfying consumer needs and recognizing high-level services by strengthening bonds [14,56]. In previous studies, customers who had trust in excellent services showed an emotional attachment; they did not consider direct purchasing behavior because of this [68]. Moreover, when customers have low affective commitment levels, competitors can easily imitate the services, and there is a risk of losing customers [14,50,56]; the research results are consistent with this study. Third, service trust affected both cognitive and affective commitments but had a strong influence on cognitive commitment. Cognitive commitment had an effect on both attitudinal and behavioral loyalty, but it was confirmed that the effect on behavioral loyalty was higher. In conclusion, service trust has a high effect on cognitive commitment, which, in turn, has a strong effect on behavioral loyalty. In terms of marketing, commitment means preferring the exchange partner or a state of attachment, feeling a sense of identity, and having a lack of alternatives. In previous studies, it has been argued that two dimensions of commitment are necessary, such as affective commitment, which is a state of attachment that shows a preference or homogeneous feeling towards a partner and perceives the need to maintain relationships because transition costs are predicted [17]. In this study, the roles of the two types of commitment were clearly identified.

*5.2. Managerial Implications*

The following practical implications can be provided through the results of this study. First, in order to increase the trust of target customers in hotel services, it is necessary to manage the service quality that can affect the evaluation of the service exchange process and results and the behavior of front-line employees interacting with customers.

Specifically, the level of service trust can be increased by reinforcing customer service education, such as prioritizing customer interests and sincerity and providing necessary services to front-line employees at the service point of contact so that target customers can strive for positive interaction. Specifically, by making the target customers perceive that the service provided by the hotel is excellent overall, the level of service quality is raised; at the same time, by strengthening the education of front-line employees who interact with customers at service points, customers can encounter service employees who are sincere and reliable. By raising the level of trust in front-line employees by creating a perception that they are sincere and reliable and consider the interests of customers their top priority, trust in the overall service can be improved. Second, it is necessary to manage customer relationships by distinguishing in terms of cognitive and affective dimensions the cases in which customers with high trust in hotel services become committed, in other words, targeting highly profitable customers after classifying them into customers with cognitive and affective inclinations according to their tendencies. There is a need to provide a comfortable service that creates a feeling of emotional attachment and a strong sense of belonging. After classifying customers with cognitive inclinations and affective inclinations, various benefits are provided to the customers with cognitive inclinations while inducing customers with affective inclinations to commit themselves by providing a pleasant and comfortable service. Third, cognitively committed customers have a high tendency to directly show behavioral loyalty, whereas affectively committed customers show loyalty (such as purchasing behavior) indirectly after mediating attitudinal loyalty. Therefore, after targeting highly profitable customers, we classify them into customers with cognitive commitment and affective commitment according to their tendencies and provide various benefits to cognitive commitment customers to induce them to commit themselves to the service, which directly leads to purchasing behavior. For affective commitment customers, it is necessary to provide a pleasant and comfortable service for them to become emotionally attached and feel a strong sense of belonging and to increase the level of attachment to the service to psychologically induce purchase behavior.

### 5.3. Research Limitations and Future Research

The significance of this study is that it presents the factors of service trust of upscale hotel services, as perceived by customers, in two dimensions: the company-related factors of perceived service quality and service satisfaction and trust in front-line employees who actually interact at the service contact point. In addition, the study is meaningful in that the relationship between trust and commitment and customer loyalty was specifically conceptualized into cognitive commitment, affective commitment, attitudinal loyalty, and behavioral loyalty, and the relationship between them was precisely analyzed. The significance of this study lies in the fact that it has identified the role of the relationship path in the process of commitment. However, this study has limitations in generalization because it is limited to only Chinese upscale hotel service users. Human behavior can be different depending on cultural and geographical differences; therefore, in future studies, we propose a comparative study by classifying differences between countries or cultures. Additionally, satisfaction as a customer's overall psychological state [69], and attitude as a psychological tendency can identify customers' future behavior in the hotel industry [70]. In future research, identifying the antecedents of service attitude or service satisfaction and how these constructs form service loyalty will also be meaningful.

**Funding:** This research received no external funding.

**Institutional Review Board Statement:** No applicable.

**Informed Consent Statement:** No applicable.

**Data Availability Statement:** No applicable.

**Conflicts of Interest:** The authors declare no conflict of interest.

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
