# Peer review of "Service Trust and Customer Loyalty in China’s Hotel Services: The Causal Role of Commitment"

_sustainability, doi:10.3390/su14138213_

Round 1

Reviewer 1 Report

It is a very complete and detailed paper, the topic analyzed is important in the field of hospitality and hotel management and the methodology used is well established in the area.

It only remains to review some minor details, such as the title of Table 4 (line 480).

Author Response

Point 1: It is a very complete and detailed paper, the topic analyzed is important in the field of hospitality and hotel management and the methodology used is well established in the area.

Response 1: Thank for evaluating this study as a meaningful study.

Point 2: It only remains to review some minor details, such as the title of Table 4 (line 480).

Response 2: The title was changed to Table 4. Result of Discriminant Validity Analysis.

Reviewer 2 Report

The paper is interesting, the topic is challenging, promising inspiring openings, useful both for theory and practice.

Methodology and approaches are interesting and systematic.

However, there are some issues that need to be solved, part of them to improve the presentation and fluency of the work

The title is long and complicated. Of course, it can stay as it is (it is the authors' choice), but a more concise and suggestive form would help a lot. Anyway, the keywords (also numerous) resume, redundantly, the main elements of the research.

The abstract is incomplete and uneven. An abstract is more than justifying the purpose and meaning of the paper but also to suggest the context and motivations of the research.

The wording is confusing:

-         See Rows 29-32 In particular, in the case of a service industry that is based on face-to-face contact with customers, such as hotel services, which has caused enormous economic losses, efforts are being made in various marketing activities to increase the trust and loyalty of the service to target customers to recover

Who / what caused enormous losses? Service industry ??

-         See Row 454  The services used were Room ??? (accommodation??)

Regarding Respondent’s Demographics and Data:

Incomes, please specify the currency?

The variables chosen are quite curious and methodologically unclear anyway:

- what kind of occupation is "housewife"?

- what is the difference between an official and an employee?

- self-employed includes entrepreneurs, business owners?

What is the purpose of collecting this data, to what extent do the selected characteristics (gender, age, education, incomes, occupation) influence the outcomes?

Interpretation of results (Ch.5. Discussion and Conclusions), although detailed and extensive, is expressed in complicated, often confusing language.

see for example rows  564-567 Third, service trust affected both cognitive and affective commitment, but had a high influence on cognitive commitment, and cognitive commitment had an effect on both attitudinal and behavioral loyalty, but it was confirmed that the effect on behavioral loyalty was higher

The paper needs proof-reading and a serios revision in layout and aspect - letters and numbers attached, extra spaces between words, different and inconsistent formatting (see hypotheses)

Although the paper aims at ambitious goals, both theoretical and practical, using consistent data and a suitable methodology, the wording, clarity of ideas and logic of sentences, is sometimes deficient. We recommend a serious revision of the text in order to reach acceptable publication standards.

Reviewer 3 Report

This research addresses an important and timely topic. The research focus on the antecedents and consequences of service trust in China’s hotel services. The research highlight the causal role of both cognitive commitment and affective commitment as well as the moderating effect of length of relationship. Overall, this is a quality research, which worth publication. Here are some suggestions for improvement:

·        I suggest revising the title of the paper. The authors talked about antecedents and consequences of trust in hotel service. However, the research did not consider neither all antecedents not all consequences. Hence, the title should focus only on the major issue of the research.

·        In line 15, the author mentioned, “a face-to-face interview was conducted on 303 customers who experienced Upscale hotel”. I think the author meant a self-administered questionnaire not a face-to-face interview. Please double check this and revise.

·        The purpose of the paper need to be clearly stated in the end of the introduction in a separate paragraph.  

·         I suggest the author split the conclusion form the discussion section.

·        The limitation of the study should highlight the direct relationship between the study construct and address this as a research opportunity

Round 2

Reviewer 2 Report

The authors carefully addressed our observations and suggestions. In the new version, the article is more suitable for publication.